# Synthesis and Characterization of Biopolyol-Based Waterborne Polyurethane Modified through Complexation with Chitosan

**DOI:** 10.3390/nano12071143

**Published:** 2022-03-29

**Authors:** Soon-Mo Choi, Sunhee Lee, Eun-Joo Shin

**Affiliations:** 1Research Institute of Cell Culture, Yeung-Nam University, 280 Daehak-ro, Gyeongsan 38541, Korea; smchoi@ynu.ac.kr; 2Department of Fashion Design, Dong-A University, 37 Nakdong-daero 550 beon-gil, Saha-gu, Busan 49315, Korea; shlee014@dau.ac.kr; 3Department of Organic Materials and Polymer Engineering, Dong-A University, 37 Nakdong-daero 550 beon-gil, Saha-gu, Busan 49315, Korea

**Keywords:** waterborne polyurethane, chitosan, reinforcement

## Abstract

In this study, a series of castor oil-based anionic waterborne polyurethane (CWPU) systems, which it has been suggested may be suitable for use as green elastomers with diverse applications in films and coatings, was prepared by modified with O-carboxymethyl chitosan (CS) as not only a reinforcing filler, but a chain-extender of polyurethane prepolymer to enhance the properties of polyurethanes. Moreover, not only was the system obtained with castor oil-based polyol in the absence of a catalyst, but it was maintained with low viscosity by using acetone instead of toxic methyl ethyl ketone (MEK) during the synthesis process. The sizes, zeta potential, chemical formation, and morphology of the CWPU-CS composites had been investigated by dynamic light scattering (DLS), infrared spectroscopy (IR), and scanning electron microscopy (SEM). Moreover, the results show that the modification allows to enhance storage/loss modulus, tensile properties, thermal stability at high temperature, and biocompatibility of CWPU and CWPU/CS nanocomposites according to various contents of CS.

## 1. Introduction

Waterborne polyurethane (WPU) has become one of the most investigated materials owing to increasingly strict environmental legislation, as an excellent alternative to solvent-based polyurethane (PU) in diverse applications such as foams [1,2,3], coatings [4,5,6,7,8], adhesives [9,10,11], oil-water separation [12,13,14], and so on [15,16]. Above all, WPU contained zero or low volatile organic compounds (VOC), which are used in many industrial fields as water is used as a dispersant, which is achieved by the introduction of the internal emulsifiers and employing water as dispersant. For this reason, WPU meets the urgent need for safety as well as environmental protection by preventing combustion or explosion during preparation, storage, transportation and painting. In addition, it can be widely applied in various fields including industrial use and medical materials due to its excellent performance such as flexibility, strength, abrasion resistance, and film form ability at a normal temperature of 25 °C [17]. It was used to synthesize polyols, isocyanates, and chain extenders, etc., as raw materials in the synthesis of polyurethane resins (PUR).

Raw materials for synthesizing WPU mainly originated from fossil resources. Due to the exhaustion of petroleum resources and the concentration of environmental issues, some research on the use of renewable materials, such as starch [16,18], chitosan [19], cellulose [17,20], natural oils [2,4,21], and so on [22,23,24] to replace petroleum-based PU, are facing up to a sustainable and greenable world trend. Among them, castor oil has become progressively significant as a renewable resource for producing polyols employed in the PU industry, due to abundant availability, its low cost, and specific structure with natural active hydroxyl groups, unlike most other vegetable oils in need of chemical modification. Approximately 90% of castor oil is composed of ricinoleic acid with one hydroxyl group on every twelfth carbon, and a double bond between the ninth and tenth carbon. This well-defined chemical structure, including available hydroxyl groups, allows castor-oil to be used as polyols in PU synthesis [25]. It is also shown that its specific triglyceride structures and long hydrophobic fatty acid chains manage to enhance thermal stability and solvent resistance of the fabricating PUs [26]. Some research results have established that castor oil-based PUs provided water resistance, good thermodynamic, and mechanical performances, such as the work by Fu et al. [27] and Liang et al. [28]. Furthermore, research has considered that the chitosan with reactive amine and hydroxyl side groups results in the formation of covalent bonds with isocyanates for the modification of WPUs, and results in non-toxicity, thermal stability, biocompatibility, and corrosion resistance [29]. However, it is hardly soluble in most organic solvents, even dissolving in acidic solutions, and this obstructs its employment in many fields [19]. In this study, water-soluble chitosan enabling us to detour the restriction that was accordingly used as a chain extender, as well as a reinforcement, to expect the effect of enhancement and the crosslinking density of the composite via a reaction with the WPU prepolymer, causing the improvement of mechanical properties. Furthermore, dimethylolbutanoic acid (DMBA) is recently considered as a new generation of green and environment-friendly internal emulsifier, which can be used without introducing organic solvents and offering zero organic residues in the production of water-based polyurethane, instead of dimethylolpropionic acid (DMPA) with a high melting point, slow dissolution, long reaction time, high energy consumption, poor product performance, the need of organic solvents, big solvent residual volume and so on [26,30]. In this article, a series of nanocomposite polymers with CWPU and CS are prepared in two steps; the first step is the preparation of CWPU prepolymer, which is synthesized from castor oil as a main polyol, IPDI, and DMBA, followed by ionizing CWPU prepolymer by triethylamine (TEA). The second step is chain-extension of CWPU incorporated by CS as both a chain extender and a reinforcing filler, via a self-emulsion polymerization method. Consequentially, a series of novel bio-based CWPU/CS dispersions was synthesized by the reaction between castor oil triol, diisocyanate, chitosan, in which the isocyanate was terminated attached to the CWPU prepolymer’s end. Remarkably, any toxic chemicals were not qualified during any of the processes, including a catalyst and MEK of a viscosity controller. Compared to MEK, acetone is much superior with regard to the environmental impact, stability, and LCA ranking, and accordingly, acetone was selected as a solvent [31]. Moreover, in the solvent removal process, MEK with a vacuum at 80 °C takes longer, and accordingly, acetone is advantageous in terms of energy efficiency.

The cast films from the obtained CWPU/CS dispersions displayed superb mechanical properties and good biocompatibility. The thermoproperties and rheology properties of composites films were also examined.

## 2. Materials and Methods

### 2.1. Materials

Castor oil 100%, Isophorone diisocyanate (IPDI) 98%, 2,2-Dimethylolbutanoic acid (DMBA) 98%, Triethylamine (TEA) 99%, acetone 99.5%, 50% Glutaraldehyde aqueous solution 98%, phosphate buffer saline (PBS), 4′,6-diamino-2′-phenylindole dihydrochloride (DAPI) > 98%, and formaldehyde solution (36.5~38%) were obtained from Sigma-Aldrich (Yongin, Korea). All chemicals used were of analytical grade and all solutions were prepared with double deionized water. O-carboxymethyl (80%) chitosan was purchased from Beijing Wisapple Biotech Co., Ltd. (Beijing, China). Dulbecco’s modified Eagle’s medium (DMEM), fetal bovine serum (FBS), penicillin/streptomycin, and trypsin-EDTA, and CellTracker^TM^ Green CMFDA (5-chloromethyl derivative of fluorescein diacetate) were supplied by Invitrogen (Carlsbad, NM, USA). Human skin fibroblasts (CCD-986K) were purchased from the American Type Culture Collection (ATCC) (Manassas, VA, USA). All aqueous solutions were made with de-ionized water.

### 2.2. Synthesis of CWPU/CS Nanocomposites and Preparation of the Films

The schematic procedure to prepare CWPU modified by CS and its films according to various CS content loading of 0~2.43 g was shown in Figure 1. For the synthesis of CWPU/CS nanocomposites, firstly, castor oil (10.0 g) and DMBA (1.86 g) were put into a four-necked flask equipped with mechanical stirrer, condenser, and thermometer. The mixture was heated to 80 °C until the castor oil was evenly mixed with DMBA, and then IPDI (7.42 g) was added dropwise. The reaction was acquired under a dry nitrogen atmosphere for 2 h. Subsequently, approximately 50 mL of acetone was poured into the flask to decrease the viscosity of the pre-polymer and followed by being cooling down to 50 °C. After neutralization of the carboxylic groups of DMBA with TEA (1.72 g) for half an hour, the distilled water or chitosan aqueous solution was slowly dropped into prepolymer acetone solution under vigorous stirring with 600 rpm. After stirring for 2 h, the rpm decreases to 200 rpm and was put at room temperature overnight, the suspension volume was reduced to a WPU solid content of 10~14 wt% by rotary vacuum evaporation at 30 °C. By changing the content, chitosan over a range from 0.405 to 2.43 g were obtained, and coded as CWPU, CWPU/CS 0.405, CWPU/CS 0.81, CWPU/CS 1.62 and CWPU/CS 2.43, respectively. There was no phase separation even if centrifuged at 5000 rpm for 30 min, in order to confirm the stability of the prepared waterborne polyurethane. Moreover, after six months, no phase separation was observed, even when visually checked, so the stability was sufficiently verified. The composition contents of CWPU and CWPU/CS nanocomposites are shown in Table 1, and the overall schematic procedure to prepare the aforementioned nanocomposites and the films are represented in Figure 1. The CWPU and CWPU/CS films were prepared by casting process. These aqueous dispersions were poured on to a glass plate, which was a set square molded type by Teflon tape with a 0.32 mm thickness. Then drying was conducted in a chamber at 25 °C and 75% relative humidity for seven days, and it was progressed in oven at 80 °C for one hour and in vacuum oven at 30 °C for 24 h, respectively. The nanocomposite films with a thickness of 0.15 ± 0.02 mm were arranged for measuring mechanical properties. These samples were stored at room temperature in a desiccator containing P_2_O_5_ with 0% relative humidity (RH).

### 2.3. Cell Seeding on the Fabricated CWPU/CS Films and Cell Imaging

Human skin fibroblasts were thawed and maintained in DMEM supplemented with 10% FBS and 1% penicillin/streptomycin in a humidified atmosphere at 37 °C and 5% CO_2_ using 25 cm^2^ flasks for thirteen passages. For the experiments, human skin fibroblasts were transferred to 75 cm^2^ flasks (2 × 10^5^ cell/flask).

The collected skin fibroblasts were labeled as follows, after removing the medium and washing confluent cells with PBS. Firstly, 1 mg of CellTracker^TM^ Green CMFDA dye was dissolved in DMSO to a final concentration of 10 mM as a stock green solution. Then, 1 μL of 10 mM stock green CMFDA dye was added to 1 × 10^4^ cells in 5 mL DMEM without serum, followed by incubation at 37 °C and 5% CO_2_ atmosphere, for 30 min. The labeled human skin fibroblasts were harvested by centrifugation and supernatant was aspirated, which were gently seeded onto each sterilized film (0.15 ± 0.02 mm in thickness and 10 mm in diameter) with a density of approximately 1 × 10^3^ cells/well. The used sterilizing protocol was delineated in our previous paper [32]. The film samples with the labeled cells with green CMFDA were incubated at 37 °C in a 5% CO_2_ atmosphere, and 1.5 mL of culture media including serum was added to each well an hour later. The samples were cultured in 12 well plates for three days in the medium and then evaluated on the first day and third day to determine the cellular proliferation on the fabricated films after nuclear staining with 4,6′-diamidino-2-phenylindole (DAPI). On the first and third day, the films with the labeled cells were stained with 200 μL of DAPI working solution (200 ng/mL with 1X PBS) after incubating in 100 μL of 10% formaldehyde solution 36.5~38% for 10 min. The stained films were washed with 1X PBS, followed by scanning to observe the level of cell proliferation by fluorescence microscopy (Ti-U; Nikon, Japan).

### 2.4. Characterization

Dynamic Light Scattering analysis: The particle size and stability of the CWPU and CWPU/CS particles in the aqueous dispersion were measured by diluting to 0.5 wt% with deionized water with a dynamic light scattering (DLS) analyzer (Zetasizer Nano ZS; Malvern Instrument Co., Ltd., Malvern, UK). The sample was added to a deionized water tank the pinhole of 200 μm and measured at 25 °C.

Nuclear magnetic resonance spectroscopy: The dried CWPU and CWPU/CS nanocomposites films were dissolved into a CDCl2 (deuterated chloroform) solvent. The 1H NMR spectra of samples were recorded in a Bruker 500 MHz NMR spectrometer by using tetramethylsilane (TMS) as standard at room temperature.

Fourier transform infrared (FT-IR) spectroscopy: The chemical structure of all prepared CWPU and CWPU/CS nanocomposites were characterized by FTIR (Spectrum 100; PerkinElmer, Waltham, MA, USA) equipped with a diamond attenuated total reflectance (ATR) accessory. The ATR spectrum was acquired in transmittance mode. The spectra were recorded at room temperature over the spectral range, 4000–650 cm^−1^, with a resolution of 4 cm^−1^.

Rotary Rheometer Measurements: The viscoelastic measurements were conducted by a rotational rheometer (MCR301; Anton-Paar, Graz, Austria) with oscillatory mode with a 25 mm diameter parallel plate fixture. The dynamic storage modulus (*G’*), loss modulus (*G”*) and complex viscosity (η *) were measured as functions of angular frequency (ω) ranging from 0.1 to 100 rad/s, using a fixed strain of 1.0, respectively.

Scanning electron microscopy (SEM): The morphological features of CWPU and CWPU/CS nanocomposite films were studied by using a JSM-6400F scanning electron microscope (SEM) instrument (JEOL, Japan) at 5 kV.

Mechanical test: The mechanical performance was surveyed was estimated by Autograph tester (Instron 4201; Sidmazu, Japan), operating at a crosshead speed of 100 mm/min. Samples of 5 mm in width, 20 mm of length, and 0.15 mm in thickness for CWPU and CWPU/CS nanocomposite films were employed with a dumbbell shape.

Thermogravimetric analysis (TGA): The analysis was accomplished by using thermogravimeter (TGA Q500, TA instruments) equipment. The nanocomposite films of 10 mg were heated from 30 to 600 °C with a heating rate of 10 °C/min under a nitrogen atmosphere.

## 3. Results

### 3.1. Synthesis of CWPU and CWPU/CS Nanocomposites

Castor oil has triglyceride with ricinoleic acid contained as a hydroxy functional group, which allows chemical reactivity with the isocyanate group in IPDI and generates the urethan group of urthane prepolymer. This hydroxyl group can be associated with the crosslinking reaction and the enhancing in mechanical properties of CWPU. For the dispersion in water, DMBA is used for the ionization with TEA. The hydroxyl group of DMBA is connected with isocyanate in PU prepolymer, followed by the generation of the urethane group. When the synthesized CWPU prepolymer with CS were dispersed in water, the urea groups are formed between the isocyanate groups of IPDI in three directions. Through the stretches of castor oil branches for urethane and urea connections, intermolecular interactions between carboxyl, hydroxyl, ammonium groups of CS and urethane, protonated amine groups of the hard segments in CWPU via multiple hydrogen bond at a molecular level, as depicted schematically in Figure 1 [33]. Even though hydrogen bonding has lower bonding energy comparing to a covalent bond, it has a great influence on phase separation, degree of crystallinity, polymer viscoelastic, and mechanical properties [34].

### 3.2. Structural Characterization of CWPU and CWPU/CS Nanocomposites Dispersions by ^1^H NMR

Figure 1 shows the ^1^H-NMR spectrum for CWPU and CWPU/CS nanocomposites films using CDCl_2_ (deuterated chloroform) as the solvent. In case of CWPU, the significant chemical shift at 7.26 ppm was the characteristic peak of CDCl_2_. The chemical shifts of soft segment corresponding to castor oil are located as follows; δ (ppm) = 0.87 (f, CH_2_O), 1.255, 1.298 (o + p, CH_2_), 2.013~2.309 (l + m + n + q, CH_2_), 2.852 (k + h, CH_2_), 4.15 (g, CH_2_), 5.339 (s, CH), 5.446 (i + j, HC). The chemical shifts of isocyanate hard segments are as follows: δ (ppm) = 1.056 (x, CH_3_), 1.171 (y, CH_3_), 2.309 (v + w, CH_2_), 4.737 (t, CH N) [35]. The chemical shift of the hydrophilic ion center is shown at δ (ppm) = 1.1 (a, CH_3_) for the methylene group of DMPA [36]. In case of CWPU/CS, the ^1^H-NMR spectrum has the following additional peaks compared to the ^1^H-NMR of the CWPU: δ (ppm) = 0.873 (10, CH), 1.986(12, CH), 2.729 (13 + 14, CH_3_, OH), 2.040 (15, CH), 2.054 (16 + 17, CH, CH_2_), 4.282 (18, CH_2_), 4.138 (20, OH) [37], which are novel chemical shift characteristic peaks produced by composite of the O-carboxymethyl chitosan (CS) chitosan. In particular, the integral value of δ (ppm) = 2.729 (13 + 14, CH_3_, OH) increases with the contents of CS.

### 3.3. Particle Size and Zeta Potential of CWPU and CWPU/CS Nanocomposites Dispersions

The particle size distribution and zeta potential of CWPU and CWPU/CS nanocomposites (n = 4) in aqueous dispersions were analyzed, as seen in Figure 2A. It is known that the particle size depends on various factors such as DMBA, chain extender, and a neutralizing agent [38], the effect of CS contents on the particle size was hereby studied with a stationary DMBA and NCO: OH ratio. The particle size of CWPU/CS 2.43 was obtained with the biggest value at 526.6 nm, while the value of CWPU no chain-extended was found to be 211.7 nm. It was indicated that the particles became bigger with an increase in the CS as the chain extender. Jang et al. reported that the size distribution generally tends to an increase in accordance with the addition of a chain extender [39]. The zeta potential can be employed for analyzing the thickness of a hydrated double electric layer of CWPU emulsion particles [17]. It was confirmed that the thickness in the double layer expended with the increase in absolute value, causing an enhancement of the mechanical/chemical stabilities. The zeta potential of the fabricated nanocomposite was correspondingly raised with increasing CS contents, which is inversely proportional to the particle size. The absolute zeta potential value was 30.5 mV, which is the highest in CWPU/CS 0.405. This result seems to be attributable to the slight tangle by increased molecule weight.

### 3.4. FTIR Studies of CWPU and CWPU/CS Nanocomposites Dispersions

Figure 2B provided the FTIR spectra of CWPU and CWPU/CS nanocomposites prepared at different CS concentrations. Firstly, it could be confirmed that most of all peaks observed corresponded with the CWPU. The absence of 2290 cm^−1^ is attributable to stretching peak of NCO groups of CWPU after the chain extension with CS [40]. The overlapping of O-H and N-H stretching peak of all samples was around 3365 cm^−1^, implying that hydrogen bonds were formed between NH and C=O groups [27]. A stretching peak of a methyl group is observed at 2943 cm^−1^, and a symmetric stretched C-H bond is present at about 2875 cm^−1^. Secondly, it could be assumed that the copolymerization between NCO groups of CWPU and NH_2_ groups of CS would occur by the formation of urea linkage, and accordingly new peak at 1691 cm^−1^ appeared clearly as the CS concentration increases in CWPU [41]. The nanocomposites had two hard segments owing to the urea and urethane bonds by forming between the diisocyanate groups and NH2/OH/COOH CS groups [42]. The resultant peaks of urethane bond established in the 1746 cm^−1^ (C=O stretch), 1021–1073 cm^−1^ (C-O stretch), 1250–1265 cm^−1^ (C-N stretch) [19]. The peak at 1387 cm^−1^ was associated with the presence of OH bending of chitosan [43].

### 3.5. Steady-Shear Flow Behavior of CWPU and CWPU/CS Nanocomposites Dispersions

Figure 3 shows the results of the steady-shear rheology analysis of CWPU and CWPU/CS nanocomposites according to various CS contents. The relationship between shear stress and shear rate in fluids can be categorized as Newtonian, pseudo-plastic, Bingham plastic, Bingham, and dilatants behavior (Yongzhen et al., 2016). There is a simple linear relationship between the shear stress and the shear rate in CWPU and CWPU/CS 0.405, which is explained by nearly Newtonian behavior of the CWPU with CS, as shown in Figure 2A. While in CWPU/CS 0.81, 1.62, and 2.43, the shear rate has a non-linear relationship, explained by pseudo-plastic behavior. The shear stress of the yield point could be obtained by extrapolation in a graph, the yield stress of CWPU/CS 0.81, 1.62 and 2.43 were 1.2 × 10^−3^, 6.2 × 10^−3^, and 2.0 × 10^−2^, respectively. The yield stress was increased according to the increase in CS contents, and had a relatively significant increase in CWPU/CS 1.62 and 2.43. As an experiment, gel was formed with the CS contents that were more than 1.62. It was considered that high yield stress was mainly due to the formation of gel structure in CWPU/CS nanocomposites, and their storage and loss modulus increased with an increase in the CS contents, as shown in Figure 2B. The storage modulus was rapidly decreased with shear rate for CWPU and CWPU/CS 0.405. In response to the crosslink and the tight hydrogen bonding between CWPU and CS, the stress decreased slowly, as the CS contents increased.

### 3.6. Dynamic Oscillation Behavior of CWPU and CWPU/CS Nanocomposites Dispersions

Figure 4 provides the rheology analysis on the dynamic shear moduli of CWPU and CWPU/CS nanocomposite dispersions according to CS contents. The frequency-dependence of the dynamic complex viscosity, η *, of CWPU and CWPU/CS nanocomposite dispersions according to CS contents is shown in Figure 4A. The values of η * suddenly increased with CS contents raised. It was observed that there was shear-thinning behavior throughout the frequency range for CWPU/CS nanocomposite dispersions with the CS content of 0.405, 0.81, 1.62, and 2.43. The η * of CWPU/CS 2.43 was much higher compared to other dispersions with lower CS contents at low frequency, however, the values were close to CWPU/CS 0.81, due to the aggregation or the gel formation resulting in higher viscosity. Furthermore, Figure 4B shows the effect of the CS content on the storage modulus (*G’*) and loss modulus (*G”*) for CWPU/CS nanocomposites as a function of frequency at 25 °C. Both *G’* and *G”* increased gradually within the increase in frequency range for all samples. With the incorporation of CS contents from 0.405 to 2.43 within the nanocomposites, higher values of *G’* and *G”* were drawn due to the crosslinking and hydrogen bonds between CS and CWPU.

### 3.7. Microscopic Studies of CWPU and CWPU/CS Nanocomposites Films

To observe the effect of various CS contents in this study, the surface morphology of the fabricated nanocomposite films was examined using SEM. The SEM images of the CWPU, CWPU/CS 0.405, 0.81, 1.62, and 2.43 are shown in Figure 5A. The figure of neat CWPU showed a uniform surface without the formation of any agglomeration, while the aggregated CWPU particles were increased on the surface of the nanocomposite films as the CS contents increased. It is considered the urethane/urea interactions or strong intermolecular interaction between amine/hydroxyl groups of CS and isocyanate group of CWPU prepolymer were formed as a result of being rough in nature and generating a phase agglomeration due to the presence of CS. Even if the phenomenon is shown considerably in the magnified images of CWPU/CS 1.62 and 2.43, there is no void between agglomerations, thereby leading to enhance the thermal and mechanical performance, which will be further addressed in the following paragraph. Since the apparent permeability of CWPU/CS nanocomposite films in Figure 1 was transparent, it can be seen that the phase agglomeration due to the presence of CS was insignificant.

### 3.8. Mechanical Properties of CWPU and CWPU/CS Nanocomposites Films

The incorporation of CS had a noticeable impact on the mechanical performances of CWPU and CWPU/CS nanocomposite films. The tensile strength and elongation changes of the fabricated films (n = 4) were investigated by stress-strain curves, as shown in Figure 5B. It could be found that the flexible CWPU film possesses a relatively low Young’s modulus of 1.2 kgf/mm^2^, a tensile strength 0.48 kgf/mm^2^ and an elongation at break of 204.4%, which is similar to typical elastomeric polymers. Additionally, the CWPU/CS 0.405 and 0.81 exhibit elongation at break with 219.7 and 238.4%, respectively, which are higher than that of CWPU films, besides the stress and Young’s modulus. It shows that the increase in their strength and elongation were caused by hard and soft segments due to the addition of CS as chain extender. While the CWPU/CS 1.62 and 2.43 displayed a decrease in the value of elongation, their stress and initial modulus increased owing to increased hard segment content by generating stiff CS chain. That is, the stiff CS chains can hinder the elasticity of the resultant films caused by crosslinking and hydrogen-bonding with CWPU chains. It was concluded with the conversion form soft elastomer to hard plastics through complexation with CS.

### 3.9. The Thermal Properties of CWPU and CWPU/CS Nanocomposite Films

The thermal property of the fabricated nanocomposite films is an important parameter for their application. The TGA and DTG curves of CWPU, CS, CWPU/CS nanocomposite films with various content of CS are presented in Figure 6. Their weight loss percentage was investigated in the range from 25 to 600 °C at 10 °C/min. The CWPU film shows four step degradation, of which two stages with weight loss are due to the fracture of weak urethane bonds, and next stage at 300~400 °C and 440~460 °C are associated with the decomposition of the soft segments. The stability of thermal degradation in hard segment tends to be lower than that of a soft segment. The third stage corresponds to the remaining double bonded fatty acid degradation of castor oil at 425~500 °C [26]. Therefore, it can be concluded that the addition of CS leads to an increase the hard segments of the CWPU/CS nanocomposite, the stability at a lower temperature becomes poor with increasing CS contents, while the thermal stability at high temperatures does not change significantly.

### 3.10. Cellular Responses on CWPU and CWPU/CS Nanocomposites Films

The purpose of this studies is to explore the possibility of a use in biomedical fields. CMFDA and DAPI staining as well as SEM analysis were employed to estimate the biocompatibility of the fabricated CWPU and CWPU/CS nanocomposite films according to CS contents with 0.405, 0.81, 1.62, and 2.43. As shown in Figure 7A–H, SEM executed for three days exhibited cellular activities including adherence, proliferation of skin fibroblasts on the fabricated films. Furthermore, the fibroblasts were stained using a green CMFDA cell tracker dye before cell seeding on the films and followed by labeling with DAPI before analyzing them, to assess that the cells were incorporated within the films. Figure 7I–L displayed photographs with cellular activity information by staining viable cells labeled with a fluorescent cytoplasmic dye (CMFDA) and cell nuclei (DAPI) represented in green and blue, respectively. These fluorescent images indicated the existence of the fibroblasts on the surface of the films. The stained images on the third day confirmed that the fibroblasts were attached and proliferated. It was founded that the CWPU/CS 1.62 and 2.43 showed a relatively remarkable biocompatibility enabling them to be applied as biomaterials, which was caused by the incorporation of CS into CWPU.

## 4. Conclusions

A series of castor oil-based anionic waterborne polyurethane (CWPU) systems was successfully developed by introducing O-carboxymethyl chitosan (CS) with loading 0.405, 0.81, 1.62, and 2.43 g through chain extension reaction. CS played an important and a leading role in the fabricated nanocomposites as crosslinkers, reinforcing fillers as well as chain extender of the CWPU prepolymer to enhance the properties, such as thermal stability, mechanical properties, and biocompatibility. The shear stress of the yield point of CWPU/CS 0.81, 1.62 and 2.43 were 1.2 × 10^−3^, 6.2 × 10^−3^, and 2.0 × 10^−2^, respectively. The initial modulus of the CWPU was 1.20 kgf/mm^2^ was increased to 11.16 kgf/mm^2^ when CS was composited in the CWPU matrix. In particular, the thermal stability of CWPU/CS 1.62 and 2.43 at high temperature (above 500 °C) was excellent due to the increase in CS leading to an increase in the hard segment. As compared to neat CWPU, a series of CWPU/CS nanocomposites showed the enhancement in the properties mentioned above, which is largely responsible for not only the hydrogen bonding between CWPU and CS, but also nucleophilic addition of the secondary hydroxyl groups of CS without the use of any catalyst. In particular, CWPU/CS 1.62 and 2.45 showed an increase in their stress and initial modulus in spite of a decrease in their elongation, this phenomenon was attributed to an increment in hard segment content by generating stiff CS chain. In other words, the CS chains obstruct the elasticity of the resultant films by crosslinking and hydrogen bonding between CS and CWPU chains. It could be concluded with the conversion from soft elastomer to hard plastics through incorporation with CS. In addition, the biocompatibility of CWPU/CS was verified by cellular response and we expect its potential for use as biomaterials will be fulfilled. Furthermore, waterborne polyurethane (WPU) chain-extended by CS enabled the acceleration of its degradation, resulting in it being more environmentally friendly. Therefore, the fabricated CWPU/CS nanocomposites with biocompatibility, as well as being eco-friendly, have potential for use in biomedical, sustainability fields.

## Data Availability

Data is contained within the article.

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
