# Peer review of "Synthesis and Characterization of Biopolyol-Based Waterborne Polyurethane Modified through Complexation with Chitosan"

_nanomaterials, 2022, doi:10.3390/nano12071143_

Round 1
Reviewer 1 Report
The manuscript developed a series of castor oil-based anionic waterborne polyurethane (CWPU) synthesized by O-carboxymethyl chitosan (CS), which is act as a chain-extender of polyurethane prepolymer to enhance the properties of polyurethanes. In addition, the properties of CWPU-CS composites such as the sizes, zeta potential, the storage/loss modulus, tensile properties, thermal stability at high temperature, and biocompatibility of CWPU and CWPU/CS nanocomposites were characterized in this study. However, I don't think this study has obvious innovation in research ideas and methods, and the experiment is not rigorous. Moreover, it is unclearly directed in application of the composites. Therefore, I think this manuscript can be consider for publication in Nanomaterials after the following major revisions were done well.
- The author should refine the highlights including that was mentioned and emphasized in abstract that “Moreover, not only was the system obtained with castor oil-based polyol in the absence of a catalyst, but it was maintained with low viscosity by using acetone instead of toxic methyl ethyl ketone (MEK) during the synthesis process.”. Besides, I don't think this study has obvious innovation in using acetone as a solvent due to that the acetone is one of the most common solvents for the synthesis of waterborne polyurethane.
- Without being characterized by using 1H-NMR spectroscopy or other characterization methods, this study cannot rigorously prove that all CMCS (as the Table 1 and Scheme 1) successfully introduced into the polyurethane molecular chain. Please supplement sufficient structure characterization.
- Furthermore, without performing post-processing operation such as dialysis or washing, the residual amount of IPDI, DMBA, TEA, castor oil and acetone in composites need to tested and assessed.
- Additionally, just three days of non-control-group cell experiment could not support the conclusion that the composites have good biocompatibility.
Author Response
Response to Reviewer #1
The manuscript developed a series of castor oil-based anionic waterborne polyurethane (CWPU) synthesized by O-carboxymethyl chitosan (CS), which is act as a chain-extender of polyurethane prepolymer to enhance the properties of polyurethanes. In addition, the properties of CWPU-CS composites such as the sizes, zeta potential, the storage/loss modulus, tensile properties, thermal stability at high temperature, and biocompatibility of CWPU and CWPU/CS nanocomposites were characterized in this study. However, I don't think this study has obvious innovation in research ideas and methods, and the experiment is not rigorous. Moreover, it is unclearly directed in application of the composites. Therefore, I think this manuscript can be consider for publication in Nanomaterials after the following major revisions were done well.
Comment 1: The author should refine the highlights including that was mentioned and emphasized in abstract that “Moreover, not only was the system obtained with castor oil-based polyol in the absence of a catalyst, but it was maintained with low viscosity by using acetone instead of toxic methyl ethyl ketone (MEK) during the synthesis process.”. Besides, I don't think this study has obvious innovation in using acetone as a solvent due to that the acetone is one of the most common solvents for the synthesis of waterborne polyurethane.
→ Most of the solvents used for the synthesis of 100% castor oil-based waterborne polyurethane are MEK, and there are few examples of successful synthesis even using acetone. When polymerization using castor oil, if the viscosity is controlled with acetone, which is less soluble than MEK, gelation occurs in most cases depending on the timing or amount of addition, therefore a lot of technical know-how is required for the synthesis experiment. Moreover, in the solvent removal process, acetone can be easily removed with vacuum at room temperature, however MEK has to be removed with a vacuum at 80°C and takes longer, accordingly acetone is advantageous in terms of energy efficiency.
Comment 2: Without being characterized by using 1H-NMR spectroscopy or other characterization methods, this study cannot rigorously prove that all CMCS (as the Table 1 and Scheme 1) successfully introduced into the polyurethane molecular chain. Please supplement sufficient structure characterization.
→ Thank you very much for your valuable advice. So, we added the result and discussion of 1H-NMR spectroscopy according to the reviewer’s indication, as below.
3.2. Structural characterization of CWPU and CWPU/CS nanocomposites dispersions by 1H NMR
Figure shows the 1H-NMR spectrum for CWPU and CWPU/CS nanocomposites using CDCl2 (deuterated chloroform) as the solvent. In case of CWPU, the significant chemical shift at 7.26 ppm was the characteristic peak of CDCl2. The chemical shifts of soft segment corresponding to castor oil are located as follow; δ (ppm)=0.87 (f, CH2O), 1.255, 1.298 (o+p, CH2), 2.013~2.309 (l+m+n+q, CH2), 2.852 (k+h, CH2), 4.15 (g, CH2), 5.339 (s, CH), 5.446 (i+j, HC). The chemical shifts of isocyanate hard segments are as follow: δ (ppm) = 1.056 (x, CH3), 1.171 (y, CH3), 2.309 (v+w, CH2), 4.737 (t, CH N) [34]. The chemical shift of the hydrophilic ion center is shown at δ (ppm) = 1.1 (a, CH3) for the methylene group of DMPA [35]. In case of CWPU/CS, the 1H-NMR spectrum has the following additional peaks compare to the 1H-NMR of the CWPU: δ (ppm) =0.873 (10, CH), 1.986(12, CH ), 2.729 (13+14, CH3, OH ), 2.040 (15, CH ), 2.054 (16+17, CH, CH2), 4.282 (18, CH2), 4.138 (20, OH) [36], which are novel chemical shift characteristic peaks produced by composite of the O-carboxymethyl chitosan (CS) chitosan. Especially, the integral value of δ (ppm) =2.729 (13+14, CH3, OH) increase with the contents of CS.
Figure 1. 1H-NMR spectra of CWPU and CWPU/CS nanocomposites with various CS contents.
Comment 3: Furthermore, without performing post-processing operation such as dialysis or washing, the residual amount of IPDI, DMBA, TEA, castor oil and acetone in composites need to tested and assessed.
→ As a result of FTIR analysis, it was confirmed that there is no isocyanate group of IPDI, and we also polymerized using DMBA and TEA with the perfect equivalent ratio for synthesis. In order to remove the solvent and acetone with the potential to remain, vacuum drying was performed at 80°C for 2 hours, after vacuum drying at room temperature for 24 hours.
Comment 4: Additionally, just three days of non-control-group cell experiment could not support the conclusion that the composites have good biocompatibility.
→ I appreciate for your indication. However, the fluorescence retention period of the CellTrackerTM Green CMFDA (green) used with DAPI was 3 days. Although it cannot be concluded that the material is excellent when used in vivo for only 3 days, it can be considered in terms of its potential. We are preparing enough to prove biocompatibility in the next study according your indication.

Reviewer 2 Report
Scheme 1 should be revised to correct the structure of the triglyceride with ricinoleic acid containing hydroxyl group in C12. The structure reported in the Scheme 1 does not shows the OH groups in the ricinoleic unit of the triglyceride,
For a complete characterization of the CPWU, the authors should also characterize the PU prepolymer by (1H and 13C)-NMR, Mass Spectrometry and SEC (Size exclusion chromatography) techniques.
I recommend checking the grammar.

Author Response
Response to Reviewer #2
This work reports the preparation a series of castor oil-based anionic waterborne polyurethane (CWPU) b nanocomposite containing different amount of O-carboxymethyl chitosan (CS), which was loaded through chain extension reaction. The sizes, zeta potential, chemical formation, and morphology of the CWPU/CS nanocomposites had been investigated by dynamic light scattering (DLS), infrared spectroscopy (IR), 21 and scanning electron microscopy (SEM). Moreover, the results show that the modification allows to enhance strorage/loss modulus, tensile properties, thermal stability at high temperature, and biocompatibility of CWPU and CWPU/CS nanocomposites according to various contents of CS. The work is sufficiently informative, it is written in a correct and legible form, the experimental are pertinent and the results obtained are well discussed. However, some revisions are necessary to publish the work in Nanomaterials.
Comment 1: Scheme 1 should be revised to correct the structure of the triglyceride with ricinoleic acid containing hydroxyl group in C12. The structure reported in the Scheme 1 does not shows the OH groups in the ricinoleic unit of the triglyceride, For a complete characterization of the CPWU, the authors should also characterize the PU prepolymer by (1H and 13C)-NMR, Mass Spectrometry and SEC (Size exclusion chromatography) techniques. I recommend checking the grammar. In conclusion, after an appropriate minor revision the work can be published in Nanomaterials.
→ We appreciate for your valuable indication. According to the indication, we modified it and added scheme 1 with the structure of triglyceride containing OH group in C12 on 5 page, as below. And we added NMR analysis in text.
Scheme 1. Schematic procedure for fabricating CWPU and CWPU/CS nanocomposites with various CS contents
Reviewer 3 Report
Interesting manuscript with a good amount of data and relevant discussions. The research presented fits well the scope of the journal. However, there are several points that must be addressed prior to further consideration.
1) Chitosan is well-known to have totally different characteristics depending on its processing/derivation and storage and source. How does that effect the final formulation and performance of the obtained materials?
2) Purity and grade for all materials used should be given under the Materials section.
3) Full chemical characterization should be provided on the nanocomposites with direct support to the chemistry outlined under Scheme 1.
4) Justification for replacing MEK with acetone should be given. Why not select a green solvent instead? The authors should refer to a general guideline on solvent selection principles (10.1016/B978-0-12-809270-5.00020-0).
5) The conditions for each step under scheme 1 should be included, which will make the scheme more information rich, and easier to understand.
6) The authors present error bars, which is appreciated, but each corresponding figure caption should mention in a sentence how these errors were derived and how many independently prepared samples were used to obtain them.
7) The use of chitosan is emerging in various fields, which should be briefly mentioned with examples (0.1039/D1GC02679H; 10.1021/acssuschemeng.1c07047; 10.1039/D1GC01799C).
8) The stability and long term integrity of the developed materials should be demonstrated in the manuscript, which is a crucial requirement for any application.
9) The potential impact and wide interest in the presented results should be better emphasized.
Author Response
Response to Reviewer #3
Interesting manuscript with a good amount of data and relevant discussions. The research presented fits well the scope of the journal. However, there are several points that must be addressed prior to further consideration.
Comment 1: Chitosan is well-known to have totally different characteristics depending on its processing/derivation and storage and source. How does that effect the final formulation and performance of the obtained materials?
→ I appreciate for your pointed question. Here, we developed a series of castor oil-based anionic WPU (CWPU) by introducing O-carboxymethyl chitosan (CS) with loading 0.405, 0.81, 1.62, and 2.43g. CS played a critical role in the fabricated CWPU as crosslinkers, reinforcing fillers as well as a chain extender of CWPU prepolymer to enhance the properties, such as thermal stability, mechanical properties, and biocompatibility. These are mentioned throughout the text.
Comment 2: Purity and grade for all materials used should be given under the Materials section.
→ We appreciate for your valuable indication. According to the indication, we modified the related sentences on page 3 of revised manuscript, as below.
“Castor oil 100%, Isophorone diisocyanate (IPDI) 98%, 2,2-Dimethylolbutanoic acid (DMBA) 98%, Triethylamine (TEA) 99%, acetone 99.5%, 50% Glutaraldehyde aqueous solution, phosphate buffer saline (PBS), 4’,6-diamino-2’-phenylindole dihydrochloride (DAPI), and formaldehyde solution (36.5~38%) were obtained from Sigma-Aldrich (Yongin, Korea). All chemicals used were of analytical grade and All solutions were prepared with double deionized water.”
Comment 3: Full chemical characterization should be provided on the nanocomposites with direct support to the chemistry outlined under Scheme 1.
→ Thank you for your valuable kind. The method of chemical characterization is provided above the scheme, and all results of full chemical characterization including NMR are provided under the scheme 1.
Comment 4: Justification for replacing MEK with acetone should be given. Why not select a green solvent instead? The authors should refer to a general guideline on solvent selection principles (10.1016/B978-0-12-809270-5.00020-0).
→ We appreciate for your valuable comment. Most of the solvents used for the synthesis of 100% castor oil-based waterborne polyurethane are MEK, and there are few examples of successful synthesis even using acetone. When polymerization using castor oil, if the viscosity is controlled with acetone, which is less soluble than MEK, gelation occurs in most cases depending on the timing or amount of addition, therefore a lot of technical know-how is required for the synthesis experiment. Moreover, in the solvent removal process, acetone can be easily removed with vacuum at room temperature, however MEK has to be removed with a vacuum at 80°C and takes longer, accordingly acetone is advantageous in terms of energy efficiency.
Comment 5: The conditions for each step under scheme 1 should be included, which will make the scheme more information rich, and easier to understand.
→ According to the indication, we modified the scheme 1, and added it, as below.
Scheme 1. Schematic procedure for fabricating CWPU and CWPU/CS nanocomposites with various CS contents
Comment 6: The authors present error bars, which is appreciated, but each corresponding figure caption should mention in a sentence how these errors were derived and how many independently prepared samples were used to obtain them.
→ We appreciate for your valuable indication. The average diameter and zeta potential graphs in Figure 2 have error bars, but it is difficult to present error bars for others (NMR, FTIR, Rheology analysis, SS curve, etc.). and at least 4 samples were used for all characterization. We added the number of used samples in corresponding figure caption, as below.
“Figure 2. (A) Average diameter and Zeta potential, and (B) FT-IR spectra of CWPU and CWPU/CS nanocomposites with various CS contents (n≧4).
Figure 5. (A) SEM images and (B) Stress-Strain curves of CWPU and CWPU/CS nanocomposites with various CS contents (n≧4).”
Comment 7: The use of chitosan is emerging in various fields, which should be briefly mentioned with examples (0.1039/D1GC02679H; 10.1021/acssuschemeng.1c07047; 10.1039/D1GC01799C).
→ We appreciate for your valuable comments. However, it seems inappropriate to mention the use of chitosan, in the flow of this article. And the use of chitosan is already known a lot, and in particular, the use of chitosan materials alone is not an important part here, so it was omitted in consideration of the overall flow.
Comment 8: The stability and long term integrity of the developed materials should be demonstrated in the manuscript, which is a crucial requirement for any application.
→ We appreciate for your valuable indication. There was no phase separation even if a centrifuge at 5000 rpm for 30 minutes, in order to confirm the stability of the prepared waterborne polyurethane. Moreover, after 6 months, no phase separation was observed even when visually checked, so the stability was sufficiently verified.
Comment 9: The potential impact and wide interest in the presented results should be better emphasized.
→ We appreciate for your valuable indication. According to the indication, we add these sentences to emphasize its potential impact, as below.
“And waterborne polyurethane (WPU) chain-extended by CS enable to accelerates its degradation, resulting in being more environmentally friendly. Therefore, the fabricated CWPU/CS nanocomposites with biocompativility as well as eco friendliness has potential for use in biomedical, sustainability fields.”

Round 2
Reviewer 3 Report
The authors either did not understand some of the questions or deliberately ignored them and gave irrelevant answers. Most of the comments still stand and much be revisited.
Comment 1: The anser has nothing to do with the comment. The problem with chitosan is that it cannot be reliable reproduced and all batches of chitosan are different, which affects the fabriaction and performance of any materials that use/incorporate chitosan. Please carefully read the original comment.
Comment 2: Glutaraldehyde aqueous solution, phosphate buffer saline (PBS), 4’,6-diamino-2’-phenylindole dihydrochloride (DAPI) ar still not specified. What is the purity? And what is the concentration of the solution?
Comment 3: C and N NMR are still missing. Peak integration for the H NMR is still missing. The structure in the top-right corner is incorrect. It seems to be a copy-pasted mirror image.
Comment 4 needs to be revisited and the general guideline referred.
Comment 5: concentration, solvent and time should also be added for all steps
In Comment 7 the authors replied that 'the use of chitosan materials alone is not an important part here' but under Comment 1 they wrote 'chitosan played a critical role in the fabricated CWPU as crosslinkers, reinforcing fillers as well as a chain extender of CWPU prepolymer to enhance the properties'. The arguments are contradictory. Indeed, chitosan is important and Comment 7 should be addressed.
Comment 8 was answered but the stability should also be discussed in detail in the manuscript.
Author Response
Response to Reviewer #3
The authors either did not understand some of the questions or deliberately ignored them and gave irrelevant answers. Most of the comments still stand and much be revisited.
Comment 1: The answer has nothing to do with the comment. The problem with chitosan is that it cannot be reliable reproduced and all batches of chitosan are different, which affects the fabrication and performance of any materials that use/incorporate chitosan. Please carefully read the original comment.
→ I appreciate for your question, but we did not bring and manufacture chitosan raw materials, we only purchase and use chitosan manufactured by a Wisapple Biotech Co., Ltd (Beiging, China). We also know well enough that chitosan has different properties depending on processing, storage, and source. The chitosan we used was water-soluble O-carboxymethyl (80%) chitosan derived from mushrooms, purchased from the company mentioned above. We cannot here prove the effect of different processing, derivation, storage, and source, we can show the effect of the ratio of “water-soluble O-carboxymethyl (80%) chitosan derived from mushrooms” on the properties of CWPU nanocomposites.
Comment 2: glutaraldehyde aqueous solution, phosphate buffer saline (PBS), 4’,6-diamino-2’phenylindole dihydrochloride (DAPI) are still not specified. What is the purity? And what is the concentration of the solution?
→ According to the indication, we added purity of DAPI on 3 page, the concentration of DAPI was already described on 4 page. The purity of PBS is not provided from any company, but we added its concentration on 4 page. The concentration and purity of glutaraldehyde aqueous solution were added on 3 page.
Castor oil 100%, Isophorone diisocyanate (IPDI) 98%, 2,2-Dimethylolbutanoic acid (DMBA) 98%, Triethylamine (TEA) 99%, acetone 99.5%, 50% Glutaraldehyde aqueous solution 98%, phosphate buffer saline (PBS), 4’,6-diamino-2’-phenylindole dihydrochloride (DAPI) >98%, and formaldehyde solution (36.5~38%) were obtained from Sigma-Aldrich (Yongin, Korea).
At 1st and 3rd day, the films with the labeled cells were stained with 200 μl of DAPI working solution (200 ng /ml with 1X PBS) after incubating in 100 μl of 10% formaldehyde solution 36.5 ~ 38% for 10 min. The stained films were washed with 1X PBS, followed by scanning to observe the level of cell proliferation by fluorescence microscopy (Ti-U ; Nikon, Japan).
Comment 3: C anc N NMR are still missing. Peak integration for the H NMR is still missing. The structure in the top-right comer is incorrect. It seems to be a copy-pasted mirror image.
→ Other reviewers also recommended H-NMR analysis, therefore we followed them and accordingly we presented in the manuscript. Now both C and N-NMR analysis are unreasonable for us. According to your valuable comments, we modified the NMR figure, as below.
Figure 1. 1H-NMR spectra of CWPU and CWPU/CS nanocomposites with various CS contents.
Comment 4: needs to be revisited and the general guideline referred
→ The reason for choosing acetone over MEK has been explained previously. I could not exactly understand your comment, so I added the reason in the use of acetone instead of MEK, according to the general guideline you referred on 2~3 page, as below.
Compared to MEK, acetone is much superior in environmental impact, stability, and LCA ranking, accordingly acetone was selected as solvent [31]. Moreover, in the solvent removal process, MEK with a vacuum at 80 °C and takes longer, accordingly acetone is advantageous in terms of energy efficiency.
Comment 5: concentration, solvent and time should also be added for all steps.
→ To show all the experimental information in the scheme, their synthetic process is too complicated, so, we described all conditions including its concentration, solvent, and time in text and Table1, on 3 page (2.2 synthesis of CWPU/CS nanocomposite and preparation of the films), as below.
2.2. Synthesis of CWPU/CS nanocomposites and preparation of the films
The schematic procedure to prepare CWPU modified by CS and its films according to various CS content loading of 0 ~ 2.43 g was shown in Scheme 1. For the synthesis of CWPU/CS nanocomposites, firstly, castor oil (10.0 g) and DMBA (1.86g) were put into a four-necked flask equipped with mechanical stirrer, condenser, and thermometer. The mixture was heated to 80 ℃ until the castor oil was evenly mixed with DMBA, and then IPDI (7.42g) was added dropwise. The reaction was acquired under a dry nitrogen atmosphere for 2 h. Subsequently, approximately 50mL of acetone was poured into the flask to decrease the viscosity of the pre-polymer and followed by cooling down 50℃. After neutralization of the carboxylic groups of DMBA with TEA (1.72 g) for half hour, the distilled water or chitosan aqueous solution was slowly dropped into prepolymer acetone solution under vigorous stirring with 600 rpm. After stirring 2h, the rpm decreases to 200 rpm and put at room temperature overnight, the suspension volume was reduced to a WPU solid content of 10 ~ 14 wt% by rotary vacuum evaporation at 30 ℃. By changing the content of chitosan over a range from 0.405 to 2.43 g were obtained, and coded as CWPU, CWPU/CS 0.405, CWPU/CS 0.81, CWPU/CS 1.62 and CWPU/CS 2.43, respectively. The composition contents of CWPU and CWPU/CS nanocomposites were shown in Table 1, and the overall schematic procedure to prepare aforementioned nanocomposites and the films was represented in Scheme 1. And the CWPU and CWPU/CS films were prepared by casting process. These aqueous dispersions were poured on glass plate, which was set square molding type by Teflon tape with 0.32 mm thickness. Then drying was conducted in a chamber at 25 ℃ and 75% relative humidity for 7 days, and it was progressed in oven at 80 ℃ for 1h and in vacuum oven at 30 ℃ for 24 h, respectively. The nanocomposite films with a thickness of 0.15±0.02 mm was arranged for measuring mechanical properties. These samples were stored at room temperature in a desiccator containing P2O5 with 0% relative humidity (RH).
Table 1. Composition of the reactants for CWPU and CWPU/CS nanocomposites with various CS contents
|
Composition |
CWPU |
CWPU /CS 0.405 |
CWPU /CS 0.81 |
CWPU /CS 1.62 |
CWPU /CS 2.43 |
Equivalent |
Molar ratio |
NCO/OH/OH |
|
Castor oil (g) |
10.00 |
10.00 |
10.00 |
10.00 |
10.00 |
0.032 |
0.021 |
0.48 |
|
IPDI (g) |
7.42 |
7.42 |
7.42 |
7.42 |
7.42 |
0.067 |
0.033 |
1 |
|
DMBA (g) |
1.86 |
1.86 |
1.86 |
1.86 |
1.86 |
0.025 |
0.013 |
0.37 |
|
TEA (g) |
1.72 |
1.72 |
1.72 |
1.72 |
1.72 |
- |
0.034 |
- |
|
CS (g) |
0 |
0.405 |
0.81 |
1.62 |
2.43 |
- |
- |
- |
|
Bio-based content (wt%) |
47.62 |
48.62 |
49.56 |
51.37 |
53.05 |
- |
- |
- |
Comment 7: The authors replied that ‘the use of chitosan materials alone is not an important part here’ but under Comment 1 they wrote ‘chitosan played a critical role in the fabricated CWPU as crosslinker, reinforcing filler as well as a chain extender of CWPU role in the fabricated CWPU as crosslinkers, reinforcing fillers as well as a chain extender of CWPU prepolymer to enhance the properties’. The arguments are contradictory. Indeed, chitosan is importane and Comment 7 should be addressed.
→ I am not saying that chitosan isn’t important, I am just saying that it is not proper to mention that the use of chitosan is emerging in various fields with examples, in terms of “the overall flow”. Of course, your suggestion may be beneficial in some paper. However, I do not want to add inappropriate content to the flow of my article. Please understand this.
Comment 8: the stability should also be discussed in detail in the manuscript.
→ We appreciate for your valuable comments. We discussed it in detail on 3 page, as below.
~~ By changing the content of chitosan over a range from 0.405 to 2.43 g were obtained, and coded as CWPU, CWPU/CS 0.405, CWPU/CS 0.81, CWPU/CS 1.62 and CWPU/CS 2.43, respectively. There was no phase separation even if a centrifuge at 5000 rpm for 30 minutes, in order to confirm the stability of the prepared waterborne polyurethane. Moreover, after 6 months, no phase separation was observed even when visually checked, so the stability was sufficiently verified. The composition contents of CWPU and CWPU/CS nanocomposites were shown in Table 1, and the overall schematic procedure to prepare aforementioned nanocomposites and the films was represented in Scheme 1. ~~
